# The Optimal Noise in Noise-Contrastive Learning Is Not What You Think

Omar Chehab[1]          Alexandre Gramfort[1]          Aapo Hyvärinen[2]

[1]Université Paris-Saclay, Inria, CEA, Palaiseau, France
[2]Department of Computer Science, University of Helsinki, Finland

## Abstract

Learning a parametric model of a data distribution is a well-known statistical problem that has seen renewed interest as it is brought to scale in deep learning. Framing the problem as a self-supervised task, where data samples are discriminated from noise samples, is at the core of state-of-the-art methods, beginning with Noise-Contrastive Estimation (NCE). Yet, such contrastive learning requires a good noise distribution, which is hard to specify; domain-specific heuristics are therefore widely used. While a comprehensive theory is missing, it is widely assumed that the optimal noise should in practice be made equal to the data, both in distribution and proportion; this setting underlies Generative Adversarial Networks (GANs) in particular. Here, we empirically and theoretically challenge this assumption on the optimal noise. We show that deviating from this assumption can actually lead to better statistical estimators, in terms of asymptotic variance. In particular, the optimal noise distribution is different from the data's and even from a different family.

## 1 INTRODUCTION

Learning a parametric model of a data distribution is at the core of statistics and machine learning. Once a model is learnt, it can be used to generate new data, to evaluate the likelihood of existing data, or be introspected for meaningful structure such as conditional dependencies between its features. Among an arsenal of statistical methods developed for this problem, Maximum-Likelihood Estimation (MLE) has stood out as the go-to method: given data samples, it evaluates a model's likelihood to have generated them and retains the best fit. However, MLE is limited by the fact that the parametric model has to be properly normalized, which

may not be computationally feasible.

In recent years, an alternative has emerged in the form of Noise-Contrastive Estimation (NCE) [Gutmann and Hyvärinen, 2012]: given data samples, it generates noise samples and trains a discriminator to learn the data distribution by constrast. Its supervised formulation, as a binary prediction task, is simple to understand and easy to implement. In fact, NCE can be seen as one of the first and most fundamental methods of *self-supervised* learning, which has seen a huge increase of interest recently [van den Oord et al., 2018, Chen et al., 2020].

Crucially, NCE can handle unnormalized, i.e. energy-based, models. It has shown remarkable success in Natural Language Processing [Mnih and Teh, 2012, Mikolov et al., 2013] and has spearheaded an entire family of contrastive methods [Pihlaja et al., 2010, Gutmann and ichiro Hirayama, 2011, Menon and Ong, 2016, Ma and Collins, 2018, Goodfellow et al., 2014, van den Oord et al., 2018].

While MLE is known to be optimal in the asymptotic limit of infinite samples, NCE is a popular choice in practice due to its computational advantage. In fact, NCE outperforms Monte Carlo Maximum Likelihood (MLE-MC) [Riou-Durand and Chopin, 2018] - an MLE estimation procedure where normalization is performed by importance sampling.

Nevertheless, NCE's performance is however dependent on two hyperparameters: the choice of noise distribution and the noise-data ratio (or, proportion of noise samples) [Gutmann and Hyvärinen, 2012]. A natural question follows: what is the optimal choice of noise distribution, and proportion of noise (or, noise-data ratio) for learning the data distribution? There are many heuristics for choosing the noise distribution and ratio in the NCE setting. Conventional wisdom in the related setting of GANs and variants [Goodfellow et al., 2014, Gao et al., 2020] is to set both the proportion and the distribution of noise to be equal to those of the data. The underlying assumption is a game-theoretic notion of optimality: the task of discriminating data from

*Accepted for the 38th Conference on Uncertainty in Artificial Intelligence* (UAI 2022).

noise is hardest, and therefore most "rewarding", when noise and data are virtually indistinguishable. The noise would then be optimal when the discriminator is no longer able to distinguish noise samples from data samples.

However, such an adversarial form of training where a noise generator aims to fool the discriminator suffers from instability and mode-collapse [Dieng et al., 2019, Lorraine et al., 2022]. Furthermore, while the above assumptions (optimal noise equals data) have been supported by numerous empirical successes, it is not clear whether such a choice of noise (distribution and ratio) achieves optimality from a statistical estimation viewpoint. Since NCE is fundamentally motivated by parameter estimation, the optimization of hyperparameters should logically be based on that same framework.

In this work, we propose a principled approach for choosing the optimal noise distribution and ratio while challenging, both theoretically and empirically, the current practice. In particular, we make the following claims that challenge conventional wisdom:

1. The optimal noise distribution is not the data distribution; in fact, it is of a very different family than the model family.

2. The optimal noise proportion is generally not 50%; the optimal noise-data ratio is not one.

The paper is organized as follows. First, we present NCE and related works in Section 2, as well as the theoretical framework of asymptotic MSE that we use to optimize the NCE estimator. We start Section 3 by empirically showing that the optimal noise distribution is not the data distribution. Our main theoretical results describing the optimal noise distribution are in Section 3.2. Specifically, we analytically provide the optimal noise for NCE in two interesting limits, and numerically verify how optimal that optimal noise remains outside these limits. We further show empirically that the optimal noise proportion is not 50% either. Finally we discuss the limitations of this work in Section 4 and conclude in Section 5.

**Notation** We denote with $p_d$ a data distribution, $p_n$ a noise distribution, and $(p_{\boldsymbol{\theta}})_{\boldsymbol{\theta} \in \Theta}$ a parametric family of distributions assumed to contain the data distribution $p_d = p_{\boldsymbol{\theta}^*}$. All distributions are normalized, meaning that the NCE estimator does not consider the normalizing constant as a parameter to be estimated to simplify the analysis: in this setup, NCE can be fairly compared to MLE and the Cramer-Rao bound is well-defined and applicable. The logistic function is denoted by $\sigma(x)$. We will denote by $\nu$ the ratio between the number of noise samples and data samples: $\nu = T_n/T_d$. The notation $\langle x, y \rangle_{\boldsymbol{A}} := \langle x, \boldsymbol{A}y \rangle$ refers to the inner product with metric $\boldsymbol{A}$. The induced norm is $\|\boldsymbol{x}\|_{\boldsymbol{A}} := \|\boldsymbol{A}^{\frac{1}{2}}\boldsymbol{x}\|$.

## 2 BACKGROUND

### 2.1 DEFINITION OF NCE

Noise-Contrastive Estimation consists in approximating a data distribution $p_d$ by training a discriminator $D(\boldsymbol{x})$ to distinguish data samples $(\boldsymbol{x}_i)_{i \in [1, T_d]} \sim p_d$ from noise samples $(\boldsymbol{x}_i)_{i \in [1, T_n]} \sim p_n$ [Gutmann and Hyvärinen, 2012]. This defines a binary task where $Y = 1$ is the data label and $Y = 0$ is the noise label. The discriminator is optimal when it equals the (Bayes) posterior

$$D(\boldsymbol{x}) = P(Y = 1|X) = \sigma\left(\frac{p_d(\boldsymbol{x})}{\nu p_n(\boldsymbol{x})}\right) \qquad (1)$$

i.e. when it learns the density-ratio $\frac{p_d}{p_n}$ [Gutmann and Hyvärinen, 2012, Mohamed and Lakshminarayanan, 2016]. The basic idea in NCE is that replacing in the ratio the data distribution by $p_{\boldsymbol{\theta}}$ and optimizing a discriminator with respect to $\theta$, yields a useful estimator $\hat{\boldsymbol{\theta}}_{\mathrm{NCE}}$ because at the optimum, the model density has to then equal the data density.

Importantly, there is no need for the model to be normalized; the normalization constant (partition function) can be input as an extra parameter, in stark contrast to MLE.

### 2.2 ASYMPTOTIC ANALYSIS

We consider here a very well-known framework to analyze the statistical performance of an estimator. Fundamentally, we are interested in the Mean-Squared Error (MSE), generally defined as

$$\mathbb{E}_{\boldsymbol{\theta}}[(\hat{\boldsymbol{\theta}} - \boldsymbol{\theta})^2] = \mathrm{Var}_{\boldsymbol{\theta}}(\hat{\boldsymbol{\theta}}) + \mathrm{Bias}_{\boldsymbol{\theta}}(\hat{\boldsymbol{\theta}}, \boldsymbol{\theta})^2$$

It can mainly be analyzed in the asymptotic regime, with the number of data points $T_d$ being very large. For (asymptotically) unbiased estimators, the estimator's statistical performance is in fact completely characterized by its asymptotic variance (or rather, covariance matrix) because the bias squared is of a lower order for such estimators. The asymptotic variance is classically defined as

$$\boldsymbol{\Sigma} = \lim_{T_d \to \infty} T_d \, \mathbb{E}_{\boldsymbol{\theta}}[(\hat{\boldsymbol{\theta}} - \mathbb{E}_{\boldsymbol{\theta}}[\hat{\boldsymbol{\theta}}])(\hat{\boldsymbol{\theta}} - \mathbb{E}_{\boldsymbol{\theta}}[\hat{\boldsymbol{\theta}}])^{\top}] \quad (2)$$

where the estimator is evaluated for each sample size $T_d$ separately. Thus, we use the asymptotic variance to compute an asymptotic approximation of the total Mean-Squared Error which we define as

$$\mathrm{MSE} = \frac{1}{T_d}\mathrm{tr}(\boldsymbol{\Sigma}) \ . \qquad (3)$$

In the following, we talk about MSE to avoid any confusion regarding the role of bias: we emphasize that the MSE is given by the asymptotic variance since the bias squared is of a lower order (for consistent estimators, and under

some technical constraints). Furthermore, the MSE is always defined in the asymptotic sense as in Eqs. (2) and (3).

When considering normalized distributions, classical statistical theory tells us that the best attainable MSE (among unbiased estimators) is the Cramer-Rao bound, achieved by Maximum-Likelihood Estimation (MLE). This provides a useful baseline, and implies that $\mathrm{MSE_{NCE}} \geq \mathrm{MSE_{MLE}}$ necessarily.

In contrast to a classical statistical framework, however, we consider here the case where the bottleneck of the estimator is the computation, while data samples are abundant. This is the case in many modern machine learning applications. The computation can be taken proportional to the total number of data points used, real data and noise samples together, which we denote by $T = T_d + T_n$. Still, the same asymptotic analysis framework can be used.

An asymptotic analysis of NCE has been carried out by Gutmann and Hyvärinen [2012]. The MSE of NCE depends on three design choices (hyperparameters) of the experiment:

- the noise distribution $p_n$

- the noise-data ratio $\nu = T_n/T_d$, from which the noise proportion can be equivalently calculated

- the total number of samples $T = T_d + T_n$, corresponding here to the computational budget

Building on theorem 3 of Gutmann and Hyvärinen [2012], we can write $\mathrm{MSE_{NCE}}$ as a function of $T$ (not $T_d$) to enforce a finite computational budget, giving

$$\mathrm{MSE_{NCE}}(T, \nu, p_n) = \frac{\nu+1}{T}\mathrm{tr}(\boldsymbol{I}^{-1} - \frac{\nu+1}{\nu}(\boldsymbol{I}^{-1}\boldsymbol{m}\boldsymbol{m}^\top\boldsymbol{I}^{-1})) \quad (4)$$

where $\boldsymbol{m}$ and $\boldsymbol{I}$ are a generalized score mean and covariance, where the integrand is weighted by the term $(1 - D(\boldsymbol{x}))$ involving the optimal discriminator $D(\boldsymbol{x})$:

$$\boldsymbol{m} = \int \boldsymbol{g}(\boldsymbol{x})(1 - D(\boldsymbol{x}))p(\boldsymbol{x})d\boldsymbol{x}$$
$$\boldsymbol{I} = \int \boldsymbol{g}(\boldsymbol{x})\boldsymbol{g}(\boldsymbol{x})^\top(1 - D(\boldsymbol{x}))p(\boldsymbol{x})d\boldsymbol{x} \quad (5)$$

The (Fisher) score vector is the gradient (or derivative in one dimension) of the log of the data distribution with respect to its parameter $\boldsymbol{g}(\boldsymbol{x}) = \nabla_{\boldsymbol{\theta}} \log p_{\boldsymbol{\theta}}(\boldsymbol{x})|_{\boldsymbol{\theta}=\boldsymbol{\theta}^*}$. Its actual (without the discriminator weight term) mean is null and its covariance is the Fisher Information matrix, written as $I_F = \int \boldsymbol{g}(\boldsymbol{x})\boldsymbol{g}(\boldsymbol{x})^\top p(\boldsymbol{x})d\boldsymbol{x}$ for the rest of the paper.

The question of statistical efficiency of NCE to bridge the gap with MLE therefore becomes to optimize Eq. 4 with respect to the three hyperparameters.

## 2.3 PREVIOUS WORK

Despite some early results, choosing the best noise distribution to reduce the variance of the NCE estimator remains largely unexplored. Gutmann and Hyvärinen [2012] and Pihlaja et al. [2010] remark that setting $p_n = p_d$ offers a MSE $(1 + \frac{1}{\nu})$ times higher than the Cramer-Rao bound. Therefore, with an infinite budget $T \to \infty$, taking all samples from noise $\nu \to \infty$ brings the $\mathrm{MSE_{NCE}}$ down to the Cramer-Rao bound.

Motivated by the same goal of improving the statistical efficiency of NCE, Pihlaja et al. [2010], Gutmann and ichiro Hirayama [2011] and Uehara et al. [2018] have looked at reducing the variance of NCE. They relax the original NCE objective by writing it as an M-divergence between the distributions $p_d$ and $p_\theta$ [Uehara et al., 2018] or as a Bregman-divergence between the density ratios $\frac{p_d}{\nu p_n}$ and $\frac{p_\theta}{\nu p_n}$. Choosing a divergence boils down to the use of specific non-linearities, which when chosen for the Jensen-Shannon f-divergence leads to the NCE estimator. Pihlaja et al. [2010] numerically explore which non-linearities lead to the lowest MSE, but they explore estimators different from NCE.

More recently, Uehara et al. [2018] show that the asymptotic variance of NCE can be further reduced by using the MLE estimate of the noise parameters obtained from the noise samples, as opposed to the true noise distribution. A similar idea underlies Flow-Contrastive Estimation [Gao et al., 2020]. While this is useful in practice, it does not address the question of finding the optimal noise distribution.

When the noise distribution is fixed, it remains to optimize the noise-data ratio $\nu$ and samples budget $T$. The effect of the samples budget on the NCE estimator is clear: it scales as $\mathrm{MSE_{NCE}} \propto \frac{1}{T}$. Consequently and remarkably, the optimal noise distribution and noise-data ratio actually do not depend on the budget $T$. As for the noise-data ratio $\nu$, while Gutmann and Hyvärinen [2012] and Pihlaja et al. [2010] report that NCE reaches Cramer-Rao when both $\nu$ and $T$ tend to infinity, it is of limited practical use due to finite computational resources $T$. In the limit of finite samples, Pihlaja et al. [2010] offers numerical results touching on this matter, although it considers the noise prior is 50% which greatly simplifies the problem as the MSE here becomes linearly dependent on $\nu$.

## 3 OPTIMIZING NOISE IN NCE

In this work we aim to directly optimize the MSE of the original NCE estimator with respect to the noise distribution and noise-data ratio. Analytical optimization of the $\mathrm{MSE_{NCE}}$ with respect to the noise distribution $p_n$ or ratio $\nu$ is a difficult task: both terms appear nonlinearly within the integrands. Even in the simple case where the data follows a one-dimensional Gaussian distribution parameterized by

variance, as specified in Section 2 of the Supplementary Material, the resulting expression is intractable. This motivates the need for numerical methods.

In the following, we pursue two different strategies for finding the optimal $p_n$. Either $p_n$ can be chosen within the same parametric family as the data distribution (we use the same parametric model for simplicity) as in Section 3.1; this leads to a simple one-dimensional optimization problem (e.g. optimizing a Gaussian mean or variance $\theta$). Or one can relax this assumption and use more flexible "non-parametric" methods as in Sections 3.2 and 3.3, such as a histogram-based expression for $p_n$. In the latter case, assuming the bins of histograms are fixed, one has in practice a higher-dimensional optimization problem with one weight per histogram bin to estimate.

## 3.1 OPTIMIZATION WITHIN THE SAME PARAMETRIC FAMILY

We use here simple data distributions to illustrate the difficulty of finding the optimal distribution. We work with families of a single scalar parameter to make sure that the numerical calculations can be performed exactly.

The data distributions considered from now on are picked among three generative models with a scalar parameter:

(a) a univariate Gaussian parameterized by its mean and whose variance is fixed to 1,

(b) a univariate zero-mean Gaussian parameterized by its variance,

(c) a two-dimensional zero-mean Gaussian parameterized by correlation, i.e. the off-diagonal entries of the covariance matrix. The variables are taken standardized.

While the Gaussian distribution is simple, it is ubiquitous in generative models literature and remains a popular choice in state-of-the-art deep learning algorithms, such as Variational Auto-Encoders (VAEs). Yet, to our knowledge, it remains completely unknown to date how to design the optimal noise to infer the parameters of a Gaussian using NCE.

Assuming the same parametric distribution for the noise as for the data, Figure 1 presents the optimal noise parameter as a function of the data parameter. Details on numerical methods are explained below. For the three models above and setting $\nu = 1$, one can observe that the noise parameter systematically differs from the data parameter. They are equal only in the very special case of estimating correlation (case c) for uncorrelated variables. This means that the optimal noise distribution is not equal to the data distribution, even when the noise and the data are restricted to be in the same parametric family of distributions.

Looking more closely, one can notice that the relationship between the optimal noise parameter and the data parame-

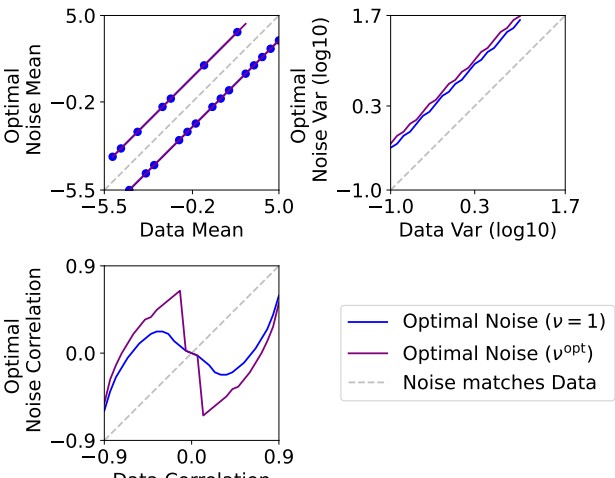

Figure 1: Relationship between the (optimal) noise parameter and the data parameter. (top left) Optimal variance in model (a) as function of the data mean. Note that the noise parameter has two symmetric local minima, given by the individual points, which are joined by a manually drawn line. (top right) Optimal variance in model (b) as function of the data variance. (bottom left) Optimal noise correlation in model (c) as a function of the data correlation.

ter highly depends on the estimation problem. For model (a), the optimal noise mean is (randomly) above or below the data mean, while at constant distance (cf. the two local minima of the MSE landscape shown in Section 1 of the Supplementary Material). For model (b), the optimal noise variance is obtained from the data variance by a scaling of 3.84. This linear relationship is coherent with the symmetry of the problem with respect to the variance parameter. Interestingly for model (c), the optimal noise parameter exhibits a nonlinear relationship to the data parameter: for a very low positive correlation between variables the noise should be negatively correlated, whereas when data variables are strongly correlated, the noise should also be positively correlated.

Having established how different the optimal parametric noise can be, a question naturally follows: what does the optimal, unconstrained noise distribution look like?

## 3.2 THEORY

While the analytical optimization of the noise model is intractable, it is possible to study some limit cases, and by means of Taylor expansions, obtain analytical results which hopefully shed some light to the general behaviour of the estimator even far away from those limits.

In what follows, we study an analytical expression for the optimal noise distribution in three limit cases: (i) when

the noise distribution is a (infinitesimal) perturbation of the data distribution $\frac{p_d}{p_n} \approx 1$; as well as when the noise proportion (ratio) is chosen so that training uses either (ii) all noise samples $\nu \to \infty$ or (iii) all data samples $\nu \to 0$. The following Theorem is proven in Section 4 of the Supplementary Material.

**Theorem 1** *In either of the following two limits:*

*(i) the noise distribution is a (infinitesimal) perturbation of the data distribution $\frac{p_d}{p_n}(\boldsymbol{x}) = 1 + \epsilon(\boldsymbol{x})$;*

*(ii) in the limit of all noise samples $\nu \to \infty$;*

*the noise distribution minimizing asymptotic MSE is*

$$p_n^{\mathrm{opt}}(\boldsymbol{x}) \propto p_d(\boldsymbol{x})\|\boldsymbol{g}(\boldsymbol{x})\|_{\boldsymbol{I}_F^{-2}} \ . \tag{6}$$

Interestingly, this is the same as the optimal noise derived by Pihlaja et al. [2010] for another, related estimator (Monte Carlo MLE with Importance Sampling). For example, in the case of estimating Gaussian variance: $p_n^{\mathrm{opt}}(x) \propto \frac{1}{\sqrt{2\pi\theta}}e^{-\frac{x^2}{2\theta}}|x^2 - \theta|$ which is highly *non-Gaussian unlike the data distribution*. Similar derivations can be easily done for the cases of Gaussian mean or correlation.

In Section 4 of the Supplementary Material, we further derive a general formula for the gap between the MSE for the typical case $p_n = p_d$ and the optimal case $p_n = p_n^{\mathrm{opt}}$. It is given by

$$\Delta\mathrm{MSE} = \frac{1}{T}\mathrm{Var}_{x \sim p_d}(\|\boldsymbol{g}(\boldsymbol{x})\|_{\boldsymbol{I}_F^{-2}}) \ . \tag{7}$$

This quantity seems to be positive for any reasonable distribution, which implies (in the all-noise limit) that the optimal noise cannot be the data distribution $p_d$. Furthermore, we can compute the gap to efficiency in the all noise limit, i.e. between $p_n = p_n^{\mathrm{opt}}$ and the Cramer-Rao lower bound $\Delta_{\mathrm{opt}}\mathrm{MSE} = \frac{1}{T}\mathbb{E}_{x \sim p_d}(\|\boldsymbol{g}(\boldsymbol{x})\|_{\boldsymbol{I}_F^{-2}})^2$.

In the third case, the limit of all data, we have the following conjecture:

**Conjecture 1** *In case (iii), the limit of all data samples $\nu \to 0$, the optimal noise distribution is such that it is all concentrated at the set of those $\boldsymbol{\xi}$ which are given by*

$$\arg\max_{\boldsymbol{\xi}} p_d(\boldsymbol{\xi})\mathrm{tr}\left((\boldsymbol{g}(\xi)\boldsymbol{g}(\xi)^\top)^{-1}\right)^{-1}$$
$$\text{s.t.} \quad \boldsymbol{g}(\xi) = \text{constant} \ . \tag{8}$$

This is typically a degenerate distribution since it is concentrated on a low-dimensional manifold, in the sense of a Dirac delta. For a scalar parameter, the function whose maxima are sought is simply $p_d(\boldsymbol{\xi})\|\boldsymbol{g}(\xi)\|^2$. An informal proof of this conjecture is given in Section 4 of the Supplementary Material. The "proof" is not quite rigorous due to the singularity of the optimal "density", which is why

we label this as conjecture only. Indeed, this closed-form formula (Eq. 8) was obtained using a Taylor expansion up to the first order. This formula is well-defined in one dimension but is challenging in higher dimensions as it involves the inversion of a rank-one matrix, which we accomplish by regularization (provided at the end of Section 4 of the Supplementary Material). While this is in apparent contradiction to having the noise distribution's support include the data distribution's, this result can be understood as a first-order approximation of what one should do with few noise data points available.

Specifically, in the case of estimating a Gaussian mean (for unit variance), the maximization in the first line of Eq. 8 yields two candidates for $p_n^{\mathrm{opt}}(x)$ to concentrate its mass on: $\delta_{-\sqrt{2}}$ and $\delta_{\sqrt{2}}$. Moreover, the second line of Eq. 8 predicts how the probability mass should be distributed to the two candidates: because they have different scores $g(-\sqrt{2}) \neq g(\sqrt{2})$, they are two distinct global minima. This is coherent with the two minima observed for the Gaussian mean in Figure 1 (top-left). Similarly, when estimating a Gaussian variance, the maximization in the first line of Eq. 8 yields candidates $\delta_{-\sqrt{5}}$ and $\delta_{\sqrt{5}}$ for $p_n^{\mathrm{opt}}(x)$. In this case however, both candidates have the same score $g(-\sqrt{5}) = g(\sqrt{5})$. The theory above does not say anything about how the probability mass should be distributed to these two points: it can be 50-50 or all on just one point. A possible solution is $p_n^{opt}(x) = \frac{1}{2}(\delta_{-\sqrt{5}} + \delta_{\sqrt{5}})$ as observed in Figure 2a. Throughout, the optimal noise distributions are highly *non-Gaussian unlike the data distribution*.

So far, we have obtained the optimal noise which minimizes the (asymptotic) estimation error $\mathbb{E}[\|\hat{\theta}_T - \theta^*\|^2] = \frac{1}{T_d}\mathrm{tr}(\Sigma)$ of NCE for the data *parameter*. However, sometimes estimating the parameter is only a means for estimating the data *distribution* — not an end in itself. We therefore consider the (asymptotic) estimation error induced by the NCE estimator $\hat{\theta}_T$ in the distribution space using the Kullback-Leibler divergence which is well-known to equal

$$\mathbb{E}[\mathcal{D}_{\mathrm{KL}}(p_d, p_{\hat{\theta}_T})] = \frac{1}{2T_d}\mathrm{tr}(\Sigma I_F) \tag{9}$$

(shown in Section 5 of the Supplementary Material). We are thus able to obtain the optimal noise for estimating the data *distribution* in cases (i), (ii) and (iii).

**Theorem 2** *In the two limit cases of Theorem 1, the noise distribution minimizing the expected Kullback-Leibler divergence is given by*

$$p_n^{\mathrm{opt}}(\boldsymbol{x}) \propto p_d(\boldsymbol{x})\|\boldsymbol{g}(\boldsymbol{x})\|_{\boldsymbol{I}_F^{-1}} \ . \tag{10}$$

In the third case, the limit of all data, we have the following conjecture:

**Conjecture 2** *In the limit of Conjecture 1 the noise distribution minimizing the expected Kullback-Leibler divergence*

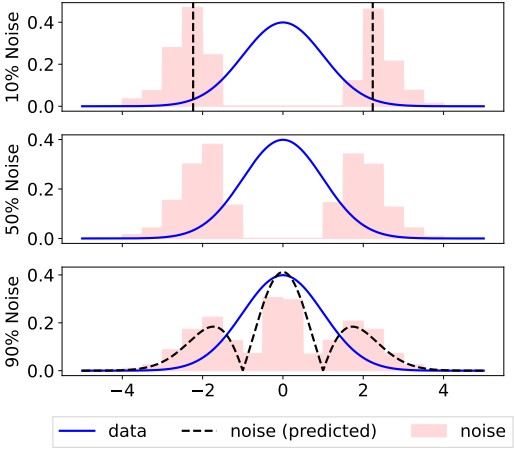

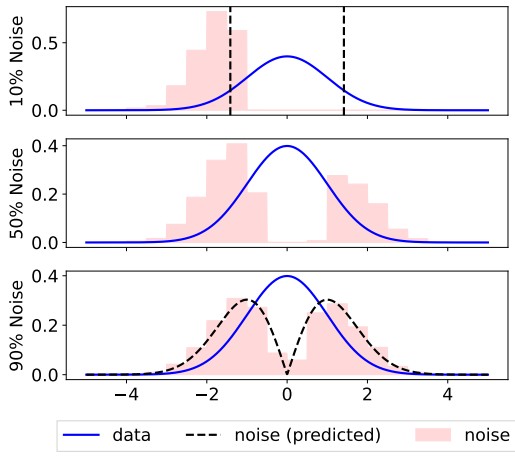

| (a) Optimal noise for model (b) (Gaussian variance). | (b) Optimal noise for model (a) (Gaussian mean). |

Figure 2: Histogram-based optimal noise distributions. Each row gives a different $\nu$ or noise proportion. The pink bars give the numerical approximations. The theoretical approximation of optimal noise is given by the dashed lines: the all-noise limit in the bottom panel, and the all-data limit in the top panel. In the top panel, the optimal noise is given by single points (Dirac masses) which are chosen symmetric for the purposes of illustration, but as explained in the text, they are two global minima in the case of Gaussian mean estimation, whereas when estimating the variance, any distribution of probability on those two points is equally optimal.

*is such that it is all concentrated at the set of those $\boldsymbol{\xi}$ which are given by*

$$\arg\max_{\boldsymbol{\xi}} p_d(\boldsymbol{\xi})\mathrm{tr}\left(\left(\boldsymbol{g}(\xi)\boldsymbol{g}(\xi)^\top\right)^{-\frac{1}{2}}\right)^{-1} \qquad (11)$$
$$\mathrm{s.t.} \quad \boldsymbol{g}(\xi) = \mathrm{constant} \ .$$

These optimal noise distributions resemble those from Theorem 1 and Conjecture 1: only the exponent on the Fisher Information matrix changes. This is predictable, as the new cost function $\frac{1}{2T_d}\mathrm{tr}(\Sigma I_F)$ is obtained by scaling with the Fisher Information matrix. More specifically, when the data parameter is scalar, the optimal noises from Theorems 1 and 2 coincide, as the Fisher Information becomes a multiplicative constant; those from Conjectures 1 and 2 do not coincide but are rather similar. The scope of this paper is to investigate the already rich case of a one-dimensional parameter, hence the following focuses on the optimal noise distributions from Theorem 1 and Conjecture 1.

### 3.3 EXPERIMENTS

We now turn to experiments to validate the theory above. Specifically, we verify our formulae for the optimal noise distribution in the all-data (Eq.8) and all-noise (Eq.6) limits, by numerically minimizing the MSE (Eq.4). Outside these limits, we show that our formulae are competitive against a parametric approach, and that the general-case optimal noise is an interpolation between both limits. We first describe numerical strategies.

**Numerical Methods** The integrals from Eq. 5 involved in evaluating the asymptotic MSE can be approximated using numerical integration (quadrature) or Monte-Carlo simulations. While both approaches lead to comparable results, quadrature is significantly faster and more precise, especially in low dimension. However, using Monte-Carlo leads to an estimate that is fully differentiable with respect to the parameters of $p_n$.

To tackle the one-dimensional parametric problem, we simply employed quadrature for evaluating the function to optimize over a dense grid and then selected the minimum. This appeared as the most computationally efficient strategy and allows for visualizing the MSE landscape reported in Section 1 of the Supplementary Material. In the multi-dimensional non-parametric case, the histogram's weights can be optimized by first-order methods using automatic differentiation.

In the following experiments, the optimization strategy consists in obtaining the gradients of the Monte-Carlo estimate using PyTorch [Paszke et al., 2019] and plugging them into a non-linear conjugate gradient scheme implemented in Scipy [Virtanen et al., 2020]. We chose the conjugate-gradient algorithm as it is deterministic (no residual asymptotic error as with SGD), and as it offered fast convergence. None of the experiments below required more than 100 iterations of conjugate-gradient. Note that for numerical precision, we had to set PyTorch's default to 64-bit floating-point precision. Our code is available at https://github.com/l-omar-chehab/nce-noise-variance.

**Results** Figure 2a shows the optimal histogram-based noise distribution for estimating the variance of a zero-mean Gaussian, together with our theoretical predictions (Theorem 1 and Conjecture 1). We can see that our theoretical predictions in the all-data and all-noise limits match numerical results. It is apparent in Figure 2a that the optimal noise places its mass where the data distribution is high, and where it varies most when $\theta^*$ changes. Furthermore, the noise distribution in the all-data limit has higher mass concentration, which also matches our predictions. Interestingly, in a case not covered by our hypotheses, when there are as many noise samples as data samples i.e. noise proportion of 50% or $\nu = 1$, the optimal noise in Figure 2a (middle) is qualitatively not very different from the limit cases of all data or all noise samples.

Figure 2b gives the same results for the estimation of a Gaussian's mean. The conclusions are similar; in this case, the optimal distributions in the two limits resemble each other even more. It is here important to take into account the indeterminacy of distributing probability mass on the two Diracs, which is coherent with initial experiments in Figure 1 as well as the MSE landscape included in Section 1 of the Supplementary Material. Figure 2b is a perfect illustration of a complex phenomenon occurring in a setup as simple as Gaussian mean estimation. Our conjecture in Eq. 8 predicts the equivalent optimal noises seen in our experiments, in Figure 1 (top-left) and Figure 2.b., where the noise concentrates its mass on either point of the set $\{-\sqrt{2}, \sqrt{2}\}$. Indeed, Eq. 8 shows that any noise which concentrates its mass on a set of points where the score is constant is (equally) optimal. So despite its approximative quality, Eq. 8 is able to explain what we observed empirically: in the all-data limit, there can be many equivalent optimal noises.

Figure 3 shows the numerically estimated optimal noise distribution for model (c) using a Gaussian correlation parameter. Here, the distributions are perhaps even more surprising than in previous figures. This can be partly understood by the extremely nonlinear dependence of the optimal noise parameter from the data parameter shown in Fig. 1.

We next ask: how robust to $\nu$ is the analytical noise we derived in these limiting cases? Figure 4 shows the Asymptotic MSE achieved by two noise models, across a range of noise proportions. The first noise model is the optimal noise in the parametric family containing the data distribution $p_n = p_\theta$, optimized for $\nu = 1$, while the second noise model is the optimal analytical noise $p_n^{\mathrm{opt}}$ derived in the all-noise limit (Eq.6). They are both compared to the Cramer-Rao lower bound. For all models (a) (b) and (c), the optimal analytical noise $p_n^{\mathrm{opt}}$ (red curve) is empirically useful even far away from the all-noise limit, and across the entire range of noise proportions. In fact, $p_n = p_n^{\mathrm{opt}}$ empirically seems a better choice than using the data distribution $p_n = p_d$, and is (quasi) uniformly equal to or better than a parametric noise $p_n = p_\theta$ optimized for $\nu = 1$.

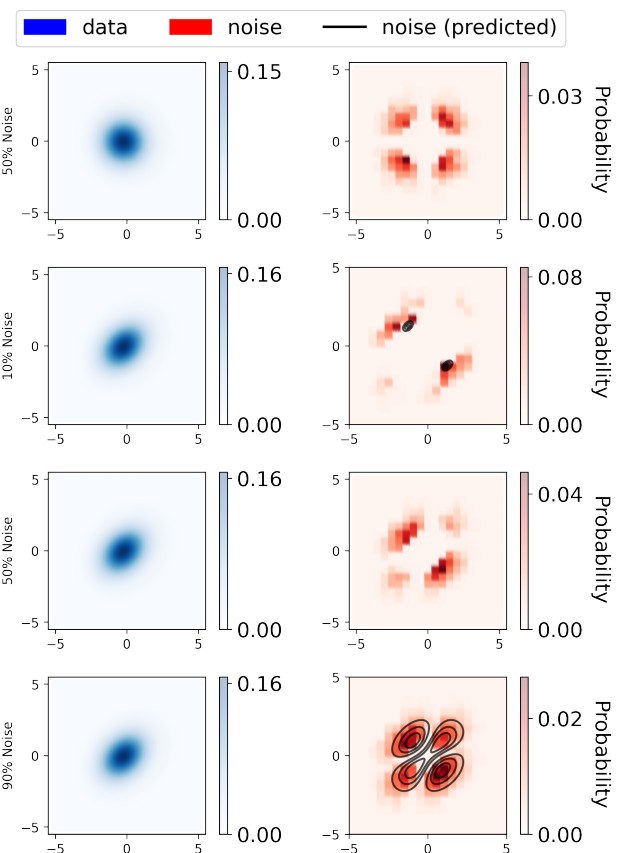

Figure 3: Optimal noise for a 2D Gaussian parameterized by correlation. 2D Gaussian with correlation 0 (top) and 0.3 (bottom three) are considered. Left panel is data density, right panel is the optimal histogram-based noise density. The theoretical approximation of optimal noise is given by the black level lines: the case of Theorem 1 the bottom panel, and the Conjecture 1 in the second panel. Here, the optimal noise in the latter limit is given by a softmax relaxation with temperature 0.01. It makes the choice of placing its mass symmetrically on the single points (Dirac masses), but as explained in the text, any distribution of probability on those two points could be equally optimal.

### 3.4 OPTIMIZING NOISE PROPORTION

Next, we consider optimization of the noise proportion. It is often heuristically assumed that having 50% noise, i.e. $\nu = 1$ is optimal. On the other hand, Pihlaja et al. [2010] provided a general analysis, although it didn't quite answer this question from a practical viewpoint; nor is it compatible with the basic NCE framework of this paper.

In the special case where $p_d = p_n$, we can actually show (see Section 3 of the Supplementary Material) that the optimal noise proportion is 50%. This is obtained for a fixed computational budget $T$, as the noise proportion varies between 0 and 1. When this constraint on the budget is relaxed,

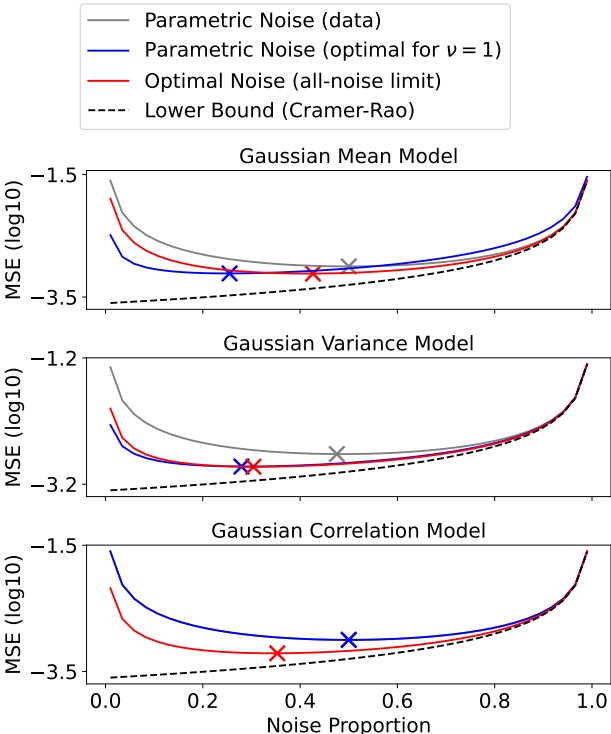

Figure 4: Asymptotic MSE vs. noise proportion. Top panel: Asymptotic MSE vs. noise proportion for model (a) with parameter mean; Middle panel: Asymptotic MSE vs. noise proportion for model (b) with parameter variance; Bottom panel: Asymptotic MSE vs. noise proportion for model (c) with parameter correlation. The parameter in "parametric noise" is the optimal parameter for $\nu = 1$, i.e. for when half the samples are noise and half are data. The "optimal noise" is the approximation given by Theorem 1.

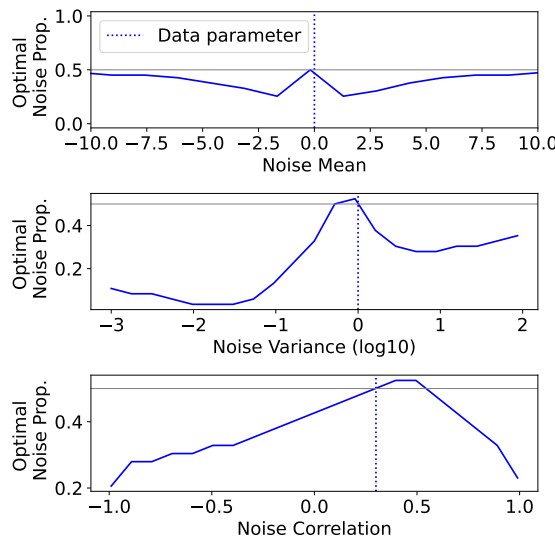

Figure 5: Optimal noise proportion against the noise parameter. Top panel for model (a), Gaussian mean; Middle panel for model (b), Gaussian variance; Bottom panel for model (c), Gaussian correlation.

for noise, it is in general less.

## 4 DISCUSSION

We have shown that choosing an optimal noise means choosing a noise distribution that is *different* to the data's. An interesting question is what implications does this have for GANs, which iteratively guide the noise distribution to *match* the data's? Both NCE and GANs in fact solve the binary task of discriminating data from noise. While the optimal discriminator for the binary task recovers the density ratio between data and noise, GANs parameterize the entire ratio (as well as the noise distribution), while NCE only parameterizes the ratio numerator. Hence they do not learn the same object, though GANs do claim inspiration from NCE [Goodfellow et al., 2014]. Moreover, of course, the goals of the two methods are completely different: GANs do not perform estimation of parameters of a statistical model but focus on the generation of data.

Nevertheless, GAN updates *have* inspired the choice of NCE noise as in Flow-Contrastive Estimation (FCE) by Gao et al. [2020], which parameterizes both the discriminator numerator and discriminator, providing a bridge between NCE and GANs. Results on FCE by Gao et al. [2020] empirically demonstrate that the choice of noise matters: NCE is made quicker by iterative noise updates *à la* GAN, presumably because setting the noise distribution equal to the data's reduces asymptotic variance compared to choosing a generic noise distribution such as the best-matching Gaussian. Noise-updates based on the optimal noise in this paper, could perhaps accelerate convergence even further, avoiding

the optimal noise proportion is $\nu \to \infty$ as in Corollary 7 and Figure 4.d. of Gutmann and Hyvärinen [2012]. The reciprocal for the theoretical result above does not hold: a noise proportion of $50\%$ does *not* ensure that the noise distribution equals the data's, as shown by counter-examples in Figures 1 and 5.

However, in the general case $p_n \neq p_d$, the optimal proportion is not 50%. We can again look at Figure 4 which analyses the MSE as a function of noise proportion for simple one-parameter families. It is not optimized, in general, at 50%, for the noise distributions considered here. In fact, the parameter of the noise distribution is here optimized for a proportion of 50%, so the results are skewed towards finding that proportion optimal, but still that is not the optimum for most cases.

A closer look at this phenomenon is given by Figure 5 which shows the optimal noise proportion as a function of a Gaussian's parameter (mean, variance, or correlation). We see that while it is 50% for when the data parameter is used

the numerical difficulties of an adversarial game while still increasing the statistical efficiency.

However, using the optimal noise distributions we present in Section 3.2 can be numerically challenging, especially when the parametric model $p_\theta$ is higher-dimensional and unnormalized (e.g. $\theta$ is a dense covariance matrix along with the normalization term as a parameter). Evaluating an optimal noise involves the Fisher score (and therefore access to the very data distribution we seek to estimate) and a Monte-Carlo method may be needed for sampling. We hope that these questions can be resolved in practice by having a relatively simple noise model which is still more statistically efficient than alternatives typically used with NCE, and whose choice is guided by our optimality results.

## 5   CONCLUSION

We studied the choice of optimal design parameters in Noise-Contrastive Estimation. These are essentially the noise distribution and the proportion of noise. We assume that the total number of data points (real data + noise) is fixed due to computational resources, and try to optimize those two hyperparameters. It is easy to show empirically that, in stark contrast to what is often assumed, the optimal noise distribution is not the same as the data distribution, thus extending the analysis by Pihlaja et al. [2010]. Our main theoretical results derive the optimal noise distribution in limit cases where either almost all samples to be classified are noise, or almost all samples are real data, or the noise distribution is an (infinitesimal) perturbation of the data distribution. The optimal noise distributions in two of these cases are different but have in common the point of emphasizing parts of the data space where the Fisher score function changes rapidly. We hope these results will improve the performance of NCE in demanding applications.

### Acknowledgements

Numerical experiments were made possible thanks to the scientific Python ecosystem: Matplotlib [Hunter, 2007], Scikit-learn [Pedregosa et al., 2011], Numpy [Harris et al., 2020], Scipy [Virtanen et al., 2020] and PyTorch [Paszke et al., 2019].

We would like to thank our reviewers whose detailed comments have helped improve this paper.

This work was supported by the French ANR-20-CHIA-0016 to Alexandre Gramfort. Aapo Hyvärinen was supported by funding from the Academy of Finland and a Fellowship from CIFAR.

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
