# OpenReview forum: "The Optimal Noise in Noise-Contrastive Learning Is Not What You Think"
_auai.org/UAI/2022/Conference — UAI 2022 Poster_

### Official Review · Reviewer_ufbJ · 2022-03-29

**Q2(1) Originality/Novelty:** 4
**Q2(2) Significance/Impact:** 4
**Q2(3) Correctness/Technical Quality:** 3
**Q2(6) Clarity Of Writing:** 3
**Q6 Overall Score:** 7
**Q8 Confidence In Your Score:** 3

**Q1 Summary And Contributions:**

This work shows the closed-form of the optimal noise distribution in noise contrastive estimation (with respect to the asymptotic variance) under a set of assumptions, which is different from the data distribution.
In the simulation study, the authors demonstrate that the numerical minimizer of the asymptotic variance indeed matches the analytical solution provided in Theorem 1.

**Q2 Assessment Of The Paper:**

More detailed information regarding each of these aspects is given below:

**Q2(4) Quality Of Experiments (Optional):**

3: Good: The experimental evaluation is adequate, and the results convincingly support the main claims.

**Q2(5) Reproducibility:**

3: Good: Key resources (e.g., proofs, code, data) are available and key details (e.g., proofs, experimental setup) are sufficiently well-described for competent researchers to confidently reproduce the main results.

**Q3 Main Strengths:**

This work demonstrates both theoretically and empirically that the optimal noise distribution in NCE is somewhat different from the data distribution, in contrast to the folklore that the good noise distribution should be data distribution.
This message brings new insight to the community such that we should take the optimal noise distribution into account more carefully in future research, and hence has a significant impact.

Although the main theorems (Theorems 1 and 2) assume the limiting conditions (i)-(iii), the authors confirm the non-limiting situations empirically in Sec. 3.3 ($\nu = 1$ or "50% noise").
Given that the experimental results under "50% noise" are similar to the limiting cases, they support that the main theorems could be applicable to a wider range of situations empirically.

I think the presentation is easy enough to follow in general.

**Q4 Main Weakness:**

While the main theorems of this paper (Theorems 1 and 2) assume relatively general situations (i)-(iii), the pilot study provided in Sec. 3.1 and the numerical simulation in Sec. 3.3 are based on specific distributional settings such as Gaussian mean estimation.
It is very instructive and illustrative to have these simulations on Gaussians but at the same time, we would like to know a little bit more empirical behaviors beyond Gaussians to understand the scope of the applicability of the results.



**Q5 Detailed Comments To The Authors:**

- In Sec. 2.3 (the 6th line in the last paragraph), "T" -> "$T$"

- In the 4th line in p.4, "we pursue two different strategies for finding the optimal $p\_n$." Explaining which sections correspond to "the same parametric family" and "non-parametric methods" would help readers to follow more easily. (I suppose the non-parametric methods are the ones described in Sec. 3.3)

- After Conjecture 1, at the end of p.5 left-column, "this result can be understood as a first-order approximation of what one should do with few noise data points available" I do not understand this sentence well. Can you elaborate a little bit more?

- At the 4th line from the end of p.5 right-column, "The scope of this paper it" -> "The scope of this paper is"

- In Figures 2 and 5, there are unnecessary commas located on the left side of the figures.

- At the beginning of Sec. 3.3, it would be helpful to make a list of what will be validated in the following experiments explicitly.

- In Sec. 3.3, the aim of the experiments of the former half is a bit unclear. In my understanding, the authors intended to confirm that the numerical minimizer of the asymptotic variance and the closed-form of the optimal noise distribution (given in Theorem 1) match. If this is the case, can you discuss how it is useful? Or, is this experiment aimed at only validating that the two solutions match?

- In Figure 2b, can you discuss why there does not exist the right peak in "10% noise"? Although the authors argue "the optimal distributions in the two limits resemble each other" in "Results", I do not think this is the case given Figure 2b.

**Q7 Justification For Your Score:**

The insights brought by this paper are technically interesting and worth advertising in the UAI community.
At the same time, the authors may consider discussing a little bit more how the insights in the optimal noise distribution can be useful and how much they are applicable beyond Gaussians.
Nevertheless, I am inclined to accept this paper given the interesting insights.

**Q9 Complying With Reviewing Instructions:**

1: Yes.

---

### Official Review · Reviewer_EaZ1 · 2022-04-10

**Q2(1) Originality/Novelty:** 4
**Q2(2) Significance/Impact:** 3
**Q2(3) Correctness/Technical Quality:** 3
**Q2(6) Clarity Of Writing:** 4
**Q6 Overall Score:** 9
**Q8 Confidence In Your Score:** 2

**Q1 Summary And Contributions:**

The authors challenge the assumptions of the optimal distribution for noise contrastive learning as well as the optimal proportion of noise to use. In both cases they find the optimal choice does not align with common conventional thought. They illustrate their point through experiments and theory.

**Q2 Assessment Of The Paper:**

More detailed information regarding each of these aspects is given below:

**Q2(4) Quality Of Experiments (Optional):**

4: Excellent: The experimental evaluation is comprehensive and the results are compelling.

**Q2(5) Reproducibility:**

4: Excellent: Key resources (e.g., proofs, code, data) are available and key details (e.g., proof sketches, experimental setup) are comprehensively described for competent researchers to confidently and easily reproduce the main results.

**Q3 Main Strengths:**

This is not an area I would describe myself as an expert and so many not be familiar with the full extent of the literature, however in my opinion this appears to be a very strong paper. They first make their points very clearly with the simple case of a one dimensional Gaussian. They build upon this with strong theoretical results supporting their hypothesis that the optimal noise is not the data distribution and the optimal proportion is not 50%. Finally they give more very nice and more detailed experimental results supporting their theory using numerical approximations. This is a paper with a very clear goal which it tackles very convincingly and with clarity.

**Q4 Main Weakness:**

I can see no clear weaknesses of the paper.

**Q5 Detailed Comments To The Authors:**

No obvious feedback to give to authors.

**Q7 Justification For Your Score:**

As mentioned this appears a very strong paper which clearly tackles a problem of importance and is likely to have a good impact on how practitioners use noise contrastive estimation. I only give a low confidence score due to my aforementioned unfamiliarity with the most up to date literature on this topic.

**Q9 Complying With Reviewing Instructions:**

1: Yes.

---

### Official Review · Reviewer_poim · 2022-04-11

**Q2(1) Originality/Novelty:** 2
**Q2(2) Significance/Impact:** 1
**Q2(3) Correctness/Technical Quality:** 3
**Q2(6) Clarity Of Writing:** 4
**Q6 Overall Score:** 6
**Q8 Confidence In Your Score:** 3

**Q1 Summary And Contributions:**

The authors focus on the properties of Noise Contrastive Estimation (NCE), for optimizing parametric models. In particular, they are interested in the optimal proportion of data between distribution data and noise data.
They present an analysis for the estimation of different parameters of small Gaussian models and conclude that the ratio of distribution to noise data should in most cases not be equal.

**Q2 Assessment Of The Paper:**

More detailed information regarding each of these aspects is given below:

**Q2(4) Quality Of Experiments (Optional):**

3: Good: The experimental evaluation is adequate, and the results convincingly support the main claims.

**Q2(5) Reproducibility:**

4: Excellent: Key resources (e.g., proofs, code, data) are available and key details (e.g., proof sketches, experimental setup) are comprehensively described for competent researchers to confidently and easily reproduce the main results.

**Q3 Main Strengths:**

The paper is very well written and technically sound. The authors present a thorough examination of the properties of NCE for the very small models introduced in section 3.1 and show different behavior for the different parameters being estimated.

The authors further demonstrate that in general, the proportion of data and noise should not be equal.


**Q4 Main Weakness:**

The authors here are tackling a very challenging problem, and indeed pose very interesting questions and provide findings for simple models. The contributions serve as a call for awareness, however, the models used by the ML community are significantly larger and the conclusion regarding the proportion of data and noise is known not to be equal.

Consider [1] where the authors use a ratio of 1:K and note that the training complexity is bounded by K.


[1] Mnih, Andriy, and Koray Kavukcuoglu. "Learning word embeddings efficiently with noise-contrastive estimation." Advances in neural information processing systems 26 (2013).

**Q5 Detailed Comments To The Authors:**

The paper is definitely welcome as a foundation for a more thorough analysis on the optimality of the hyper-parameters of NCE.

The exposition presented on the small models, follows a thorough technical discussion and provide some insights. It however does not claim to extend to other much larger models, nor provides further insights that were not previously shown for the average practitioner.

**Q7 Justification For Your Score:**

The score reflects the expected impact of the contribution. As shown before in [1], practitioners are using noise proportions already not equal and only limited by computational challenges.

**Q9 Complying With Reviewing Instructions:**

1: Yes.

---

### Official Review · Reviewer_SnuG · 2022-04-18

**Q2(1) Originality/Novelty:** 2
**Q2(2) Significance/Impact:** 3
**Q2(3) Correctness/Technical Quality:** 3
**Q2(6) Clarity Of Writing:** 3
**Q6 Overall Score:** 6
**Q8 Confidence In Your Score:** 3

**Q1 Summary And Contributions:**

Noise-Contrastive Estimation requires the design of noise distribution. However, such design is hard to specify and usually requires domain-specific heuristics. This paper empirically and theoretically studies the optimal noise for Noise-Contrastive Estimation.

**Q2 Assessment Of The Paper:**

More detailed information regarding each of these aspects is given below:

**Q2(4) Quality Of Experiments (Optional):**

2: Fair: The experimental evaluation is weak: important baselines are missing, or the results do not adequately support the main claims.

**Q2(5) Reproducibility:**

3: Good: Key resources (e.g., proofs, code, data) are available and key details (e.g., proofs, experimental setup) are sufficiently well-described for competent researchers to confidently reproduce the main results.

**Q3 Main Strengths:**

The paper is well written. The work is well motivated.

**Q4 Main Weakness:**

The experiments considered in the paper are mostly low-dimensional datasets. It would be good to consider more challenging high-dimensional datasets. It remains unclear whether the proposed approach could scale to high-dimensional data.

**Q5 Detailed Comments To The Authors:**

The experiments are discussion are mainly for 1D or 2D data distributions, it would be good to expand the discussion to more challenging high-dimensional data.

**Q7 Justification For Your Score:**

This paper considers a very interesting direction and provides theoretical analysis for the optimal noise in NCE. However, the discussion and experiments are mainly for 1D and 2D data. It remains unclear how empirically useful the proposed approach is for high-dimensional data.

**Q9 Complying With Reviewing Instructions:**

1: Yes.

---

### Decision · Program_Chairs · 2022-05-15

**Decision:**

Accept (Poster)

**Comment:**

Meta Review: The reviewers and myself agree that this is an interesting theoretical paper on characterizing the optimal (in the sense of asymptotic sample complexity) noise distribution for noise contrastive estimation. Though the results are mostly derived for simple data distributions (e.g. Gaussian), some of the results are unexpected, and along with empirical verification, suggest that they could potentially generate insights for more complicated real-life settings.